# Disability-Related Mortality Inequity in South Korea: Comparison in Terms of Seoul Metropolitan/Non-Metropolitan Areas and Income Levels

**DOI:** 10.3390/healthcare12030293

**Published:** 2024-01-23

**Authors:** Seo-Yeon Chung, Ru-Gyeom Lee, So-Youn Park, In-Hwan Oh

**Affiliations:** 1Department of Preventive Medicine, School of Medicine, Kyung Hee University, Seoul 02447, Republic of Korea; seoyeon_chung@khu.ac.kr (S.-Y.C.); lrg0227@khu.ac.kr (R.-G.L.); 2Department of Medical Education and Humanities, School of Medicine, Kyung Hee University, Seoul 02453, Republic of Korea

**Keywords:** disparities, health equity, health inequality, mortality

## Abstract

Despite the improved living standards in South Korea, people with disabilities still experience health disparities. Therefore, we analyzed differences in mortality rates among people with disabilities according to income level and residential area using representative data from the National Health Insurance Service in South Korea. Descriptive statistics and Cox proportional risk models were used to identify the risk factors for mortality affecting people with disabilities stratified by income level and residential area. Those living in non-metropolitan areas and low-income households had high mortality risks, suggesting that income level and residential area were related to mortality risk. The mortality risk of those with a high-income level was 1.534 times higher in non-metropolitan areas than in Seoul metropolitan areas (95% confidence interval [CI] = 1.44–1.63). Among people with low income living in non-metropolitan areas, the crude hazard ratios of mortality risk were 1.26 (95% CI = 1.14–1.39), 1.44 (95% CI = 1.34–1.54), and 1.39 (95% CI = 1.20–1.61) for those with brain lesions, sensory impairment (visual/hearing/speech impairment), and kidney failure, respectively. No significant differences were observed between people with autism in non-metropolitan and Seoul metropolitan areas and those with low- and high-income levels. Health issues and countermeasures are crucial to reduce mortality risk among people with disabilities.

## 1. Introduction

Inequalities in socioeconomic aspects, such as individual income, residential area, type of disability, and lack of access to healthcare services, can lead to health inequity [1,2,3,4,5,6,7]. Health inequity comprises essential living elements that determine people’s living standards and cannot be explained by a single factor. For example, social determinants of health, such as sex, race, socioeconomic status, geographic location, and access to quality health care, affect a person’s health status; if these social bases are weak, a healthy life cannot be achieved [8]. Particularly, an individual’s socioeconomic position is a strong predictor of health because economic deprivation can cause avoidable diseases and even death [9]. As inequity worsens, the health of members of this society is expected to be adversely affected.

According to data from the 2020 National Survey of Persons with Disabilities, people aged ≥65 years accounted for 43.3% of the population with disabilities in 2014, 46.6% in 2017, and 49.9% in 2020. Moreover, as of 2021, the proportion of people with disabilities was significantly higher in small and medium-sized cities and rural areas than in large cities [10]. People with disabilities comprise a large part of the population; however, they are one of the most underserved subpopulations [11]. Health disparities in people with disabilities, as reported in previous research, include differences in the incidence, prevalence, mortality, and burden of diseases, and other adverse health conditions, as well as differences in outcomes and access to healthcare services [12,13,14]. For example, the differences in health screening rates are shown according to education and economic status among the PWD group in Korea [15]. Particularly, the need for income security and healthcare services is increasing due to the prolonged coronavirus disease 2019 pandemic [16,17]. Most older adults in Korea are supported by the National Health Insurance Service (NHIS) through long-term care insurance, health insurance, and medical benefits. However, not only are their income levels, medical examination data, and characteristics of diseases not considered, but also their access to medical services is restricted due to out-of-pocket expenditure relative to their financial resources [18]. Therefore, people with disabilities should be considered targets of active health promotion and disease prevention.

Due to some limitations in daily life, people with disabilities spend more money on medical expenses and have poorer access to healthcare services than people without disabilities. Moreover, their income is often insufficient, making access to medical services difficult. People with disabilities might be marginalized from the benefits of medical care facilities and services based on their income level, which can lead to health inequity and mortality [19]. Additionally, previous studies have found a higher prevalence of disability among the poor [20,21]. Few studies have followed up on mortality risk in people with disabilities who have died [22]. Several studies have reported increased mortality risks for different disability types and severity levels, such as motor impairment or mental illness [23]. However, the differences in mortality rates among people with disabilities, based on socioeconomic and regional disparities, have not been studied yet. Therefore, when considering the theoretical underpinnings of our study, we draw upon the work of Meade et al., who developed a comprehensive conceptual framework illuminating the intersection of disability and healthcare disparities [24].

This study sought to explore the differences in mortality rates among people with disabilities according to income level and residential area using a nationally representative sample of the South Korean population, with data sourced from the NHIS. We specifically concentrated on the effect of residential areas, an aspect often discounted compared to income levels, among PWDs. Among the residential areas, we tried to understand the impact of the Seoul metropolitan area, where half of the population and 54% of medical personnel are concentrated [25]. The impact of these residential areas, especially whether they live in the Seoul metropolitan area, has not been dealt with in previous studies. We hypothesized that mortality risk would be lower for people with disabilities living in non-metropolitan areas compared with those living in Seoul metropolitan areas.

## 2. Materials and Methods

### 2.1. Data Sources

Data from the NHIS database, a South Korean single-payer public insurance system, were retrospectively analyzed. The NHIS database contains publicly available data, such as population-based medical history, death records (excluding cause of death), health examination records, and general characteristics (income bracket, BMI, smoking and drinking status, etc.), to support policymaking and academic studies [26]. In the case of the study on the cause of death, when applying for data to NHIS, death data (information on the cause of death) could be provided by Statistics Korea. The data are only accessible to researchers with pre-approved access credentials granted by the official review committee. The personal information provided by the NHIS data is deidentified to ensure that personal identification is not possible. For this study, we collected variables on socio-economic (disability type, age, sex, BMI, smoking status, drinking status, demographic location), health insurance claims (income bracket), and mortality rates from claims data that were recorded from 2019 to 2020.

### 2.2. Participants

In the NHIS claims database, patients were codified using a comprehensive disability grade code (CMPR_DSB_GRADE) or main disability type (MAIN_DSB_TYPE) code. The classification of disability types was undertaken utilizing data sourced from the NHIS. In Korea, the PWDs are defined and classified based on “the Welfare of the PWD Act” which is based on the medical model [27]. NHIS has and provides patient information classified into distinct category types, each delineated by specific codes for PWD-specific benefits. These codes encompass a spectrum of conditions, including but not limited to physical disability (‘01’), brain lesions (‘02’), visual/hearing impairments (‘03–04’), intellectual disabilities (‘06–07’), and kidney-related conditions (‘09’). The claims data of 2,711,941 were retrieved, and 1,317,012 patients who did not have health examinations during 2019–2020 or with missing values or erroneous values were excluded. Participants without missing values for the study variables were included in the final study sample, which included 1,394,929 people with disabilities (Figure 1) (physical disabilities, 771,104; brain disorders, 786,764; visual and language impairment, 363,426; intellectual/autistic disability, 69,804; renal disability, 33,330; and others, 70,564 people).

### 2.3. Statistical Analysis

Descriptive statistics were used to analyze the demographic characteristics of the participants, and the results were expressed as frequencies, with mean. The risk of mortality according to income level and area of residence was analyzed by a Cox proportional hazard model, and the results were expressed as hazard ratios (HRs), with 95% confidence intervals (CIs). The mortality rate (index) was based on the date of death attached to each non-identified patient in the NHIS database. All the variables provided by the NHIS were applied. Income level was classified into three groups based on the insurance premium level: low-income (0–6 deciles), middle-income (7–14 deciles), and high-income (15–20 deciles).

The classification of residential areas was based on Article 2 of the Enforcement Decree of the Seoul Metropolitan Area Readjustment Planning Act related to urbanization and population density. Seoul, Gyeonggi, and Incheon were categorized as metropolitan areas due to their high population density, extensive urban infrastructure, and significant economic and cultural activities. In contrast, other cities were categorized as non-metropolitan areas, considering factors such as smaller population size, lower urbanization levels, and limited economic and cultural development [28,29].

We controlled for the effects of confounding factors to minimize bias due to loss during follow-up. Analysis models were adjusted for sex (male or female), age range (20–29, 30–39, 40–49, 50–59, 60–69, 70–79, and >80 years), disability type (physical, brain lesion, visual/hearing/language, intellectual/autistic, kidney, and others), BMI (normal [18.5–24.9], underweight [<18.5], overweight [25.0–29.9], or obese [≥30] kg/m^2^), smoking status (non-smokers, former smokers, or smokers), alcohol consumption (0, 1–2, 3–4, and ≥5 days per week), and Charlson comorbidity index.

The null hypothesis is that income level and place of residence do not affect the mortality rate among PWD when other factors are corrected. The SAS Enterprise Guide tool (Base SAS version 9.4) provided by the NHIS was used for statistical analysis, and the significance level for all statistical tests was set at *p* < 0.05. This study was approved by the Institutional Review Board of Kyung Hee University (Approval No. KHSIRB-21-397[EA]).

## 3. Results

The general characteristics of participants with disabilities who died during the study period and the general characteristics of the entire population are presented in Table 1 and Appendix A, respectively. Regarding the type of disability, the majority of those who died had physical disabilities, followed by those with visual/hearing/speech disabilities, brain lesions, other disabilities, kidney failure, and intellectual disabilities. Most participants with physical disabilities who died resided in non-metropolitan areas. The proportion of older people was higher than that of younger people. In particular, the percentage of people aged >80 years who lived in a non-metropolitan area and had a high income was the highest at 51.2%. The deceased population dataset included fewer people with obesity than those with lesser weight, fewer smokers than former and non-smokers, and fewer heavy drinkers than infrequent drinkers.

Table 2 presents the results of the univariate analysis of mortality risk according to demographic characteristics. The differences in mortality risk according to income level and residential area were significant (high income/non-metropolitan areas: HR = 1.13, 95% CI = 1.09–1.17; median income/metropolitan areas: HR = 1.21, 95% CI = 1.15–1.27; median income/non-metropolitan areas: HR = 1.28, 95% CI = 1.23–1.33; low income/metropolitan areas: HR = 1.22, 95% CI = 1.16–1.27; low-income/non-metropolitan areas: HR = 1.45, 95% CI = 1.39–1.51). In the model adjusted for alcohol consumption, mortality risk was higher among men (HR = 1.68, 95% CI = 1.64–1.73) than among underweight women (HR = 2.37, 95% CI = 2.19–2.56) and former smokers (HR = 2.14, 95% CI = 2.05–2.24).

The differences in mortality risk according to the type of disability, income level, and region were analyzed in a model adjusted for age, sex, BMI, smoking status, alcohol consumption, and the Charlson comorbidity index (CCI; Table 3). Compared with high-income individuals with physical disabilities who resided in Seoul metropolitan areas, those who resided in non-metropolitan areas had a mortality risk of 1.16 (95% CI = 1.09–1.23), whereas median-income individuals who resided in metropolitan and non-metropolitan areas had risks of 1.26 (95% CI = 1.17–1.35) and 1.32 (95% CI = 1.24–1.41), respectively. Similarly, low-income individuals with disabilities who resided in metropolitan and non-metropolitan areas had risks of 1.31 (95% CI = 1.22–1.41) and 1.53 (95% CI = 1.44–1.63), respectively. Moreover, low-income participants who resided in non-metropolitan areas and had brain lesions, sensory impairment (visual/hearing/speech impairment), and kidney failure had crude HRs of mortality risk of 1.26 (95% CI = 1.14–1.39), 1.44 (95% CI = 1.34–1.54), and 1.39 (95% CI = 1.20–1.61), respectively. However, there was no significant difference in the case of autism.

## 4. Discussion

This study aimed to identify mortality risk among people with disabilities according to income level and region of residence using representative data from the NHIS. We found that the HR for mortality was higher areas after controlling for other variables, and the difference was statistically significant, even after considering the effects of income, the presence of autism, and the presence of kidney failure. The physical disability group had the greatest risk of death, regardless of the area of residence and income level.

According to the World Health Organization, more than 1 billion people worldwide currently have disabilities, accounting for approximately 15% of the population [30]. As inequities worsen, the health of members of society is also expected to be adversely affected. In particular, income inequality is associated with an increased risk of subjective health deterioration and mortality [31]. The average poverty rate of people with disabilities in South Korea is 1.6 times higher than the rates in other Organization for Economic Co-operation and Development member countries (35.6% vs. 22.1%) [32]. People with disabilities face severe deprivation not only in terms of income but also in terms of education, health, and housing. In a study on the impact of multidimensional poverty on people with disabilities in Korea, the influence of health accounted for 18.14%, and assets for 17.14% of the total impact [33].

The occurrence of diseases and disabilities cannot be attributed solely to poverty; however, in many previous studies, the likelihood of death and disability has been reported to increase as income decreases [34]. Social vulnerability is an important predictor of mortality and disability [35] because people with disabilities have high incidental costs of transportation, utilities, and daily life management [36]. And also, people with disabilities spend less time in paid work because they are vulnerable to social welfare that provides the conditions and facilities for them to participate in the productive process [37]. In addition, in the case of families with a member with a disability, the other family members may not be able to engage in adequate employment activities because they would have to take care of the member with a disability [38]. In other words, income itself impacts the onset of disabilities, and people with disabilities are more vulnerable to economic changes and more easily exposed to poverty than those without disabilities.

In the present study, there were differences in mortality risk factors among people with disabilities residing in metropolitan areas and health inequity was pronounced across these groups. Health status can be affected by the living environment and income level. Korean society is divided into urban and rural and metropolitan and non-metropolitan areas, owing to unbalanced growth between regions and unilateral regional development [39]. The difference between urban and rural areas decreased after industrialization; however, the health differences between metropolitan and non-metropolitan areas remained [40]. A health gap has also been observed between metropolitan and non-metropolitan areas. Allan et al. [41] found that people living in London, UK, were healthier than those living in other cities, similar to the results of the present study. This regional disparity can be explained by polarization due to the concentration of healthcare services in metropolitan areas, and this disparity strongly affects people with disabilities who already face physical and mental barriers to accessing healthcare.

In general, people with disabilities require more healthcare services and visit hospitals more frequently than healthy people [42]. However, the limited mobility of people with disabilities creates barriers to healthcare access, and this accessibility differs based on their location of residence. People with disabilities living in rural areas find it difficult to access healthcare services [43], and the rates of healthcare and medical deprivation among middle-aged people with disabilities living in non-metropolitan areas are high [44].

In our study, a high number of people with physical disabilities were living in non-metropolitan areas. Physical disabilities are acquired, and they are the most common type among older adults. A previous study reported that the prevalence of physical disabilities in the elderly population aged ≥65 years was 9.51%, higher than 2.55% in the total sample [45]. In addition, a study found that older adults living in the metropolitan area accounted for 15.4% and those living in non-metropolitan areas accounted for 19.5% of the total population [46]. As can be found in recent studies, the high proportion of older adults with disabilities living in non-metropolitan areas may, in part, explain the higher prevalence of physical disability in the non-capital area, as shown in the present study.

Another study showed that people with disabilities were more likely to earn insufficient salaries and experience health disparities [47], and place of residence was closely associated with visual impairment and intellectual and developmental disabilities [48]. Compared with other types of disabilities, mortality risk was particularly high for people with physical disabilities living in non-metropolitan areas and low-income households. As of 2021, people with physical disabilities accounted for 45.1% of the population with disabilities in South Korea [49], whereas in the USA, as of 2020, 13.7% of the population with disabilities had severe difficulty walking or climbing stairs [50].

People with physical disabilities have limited motor function and require continuous management and rehabilitation [51]. Moreover, they have limited access to healthcare services because of the long distance between healthcare centers and their homes and the lack of transportation services, especially in non-metropolitan areas [52]. Moreover, the high medical and other health-related expenses in low-income households can restrict their access to healthcare services [53], negatively affecting their long-term health. Although physical disability has not been found to increase mortality risk, in older adults, the association may be confounded or mediated by factors associated with mortality, such as cognitive impairment [54]. Therefore, it is necessary to review and discover related national policies that allow people with physical disabilities to receive necessary health and medical services in a timely manner, such as expanding convenience facilities and supporting medical welfare. In addition, among those with brain lesions, the mortality risk in those living in non-metropolitan areas was high for all income groups. Studies have been conducted on stroke-related mortality in the context of socioeconomic status [55,56]. Hence, to address the problems listed above, it is necessary to determine ways to improve accessibility to healthcare services through strategies such as telehealth or digital healthcare.

### 4.1. Strengths and Limitations of the Study

This study has some limitations. The proportions of people with developmental disabilities and kidney failure were relatively small; hence, they could not be analyzed. Differences in occupation were also not investigated in this study. Furthermore, 2-year data from 2019 to 2020 was used; therefore, the observation period was limited. Further investigation among cohorts is required for differences in mortality levels and causes of death according to the time of onset of disability (congenital or acquired) and duration of disability. One limitation arises from our reliance on administrative data for classifying Seoul’s metropolitan and non-metropolitan areas. While administrative data provide a valuable quantitative foundation, the nature of these data limits our ability to delve into the nuances and variations among Seoul metropolitan areas. For instance, within the Seoul metropolitan area, there exist diverse social, economic, and health-related factors that may influence health inequity.

Despite these limitations, this study has several strengths. The NHIS database covers almost the entire population; it is the only database that includes data on all medical service use of the total national population. This database is the closest to real-world data, which means big data in healthcare [57]. This study contributes valuable insights into the overall mortality risk in people with disabilities, considering income levels across Seoul’s metropolitan and non-metropolitan areas.

### 4.2. Implications and Future Directions

Based on our findings, the implications for health policy and intervention strategies are profound. Public health and welfare political policy interventions that consider both income level and geographic location are essential for addressing mortality disparities and premature death among people with disabilities. Our study highlights the unique importance of geographic location for this population, emphasizing the need for tailored healthcare policies that account for regional variations in access to medical care.

In the Republic of Korea, Physicians’ Experience from the Primary Care for People with Disabilities, a commendable initiative currently operational in 617 locations nationwide, is a significant step toward enhancing healthcare access for individuals with disabilities [15]. However, there is a notable concentration disparity—278 locations in the Seoul metropolitan area compared to 339 locations in non-metropolitan areas (the land area proportions of the metropolitan area (11.8%) and the non-metropolitan area (88.2%)). The findings of this study suggest that simply increasing the number of facilities is not enough, especially for PWD. Rather, it is important to ensure their equal distribution for the health of PWD.

Future research should focus on in-depth examinations of regional disparities, exploring the specific factors that contribute to variations in mortality risk. Comparative studies across regions can further enrich our understanding of how social, economic, and healthcare infrastructure factors interact in specific contexts. Additionally, evaluating interventions aimed at reducing both income-related and geographic barriers to healthcare access is crucial for developing effective and equitable healthcare strategies for people with disabilities.

Such studies will provide valuable insights into the dynamic nature of health disparities and inform the development of sustainable interventions. By addressing these future research directions, we can contribute to the ongoing efforts to enhance the health and well-being of people with disabilities.

## 5. Conclusions

In conclusion, mortality risk in people with disabilities was higher in those who lived in non-metropolitan areas and those with low income, compared with their respective counterparts. Our study sheds light on the intricate relationship between income, geographic location, and the mortality risk faced by individuals with disabilities. The compelling findings underscore the pivotal role of geographic location in shaping health outcomes for this vulnerable population. While income level also contributes to mortality risk, the synergy between income and geographic location unveils a nuanced interplay. The combination of residing in non-metropolitan areas amplifies the disparities, indicating that interventions addressing both factors are crucial for effective healthcare policy. And our study underscores the pervasive health inequities faced by people with disabilities, emphasizing the urgent need for targeted interventions. As we delve into the layers of these disparities, it becomes evident that a status quo approach is insufficient. The urgency of the situation necessitates a paradigm shift in our approach to healthcare for this vulnerable population (e.g., telehealthcare service). Furthermore, our findings serve as a call to action for additional research focusing on the development and evaluation of health promotion programs, as well as the assessment of healthcare and welfare policies. Future studies should aim to provide evidence-based solutions to prevent premature deaths in this vulnerable population.

## Figures and Tables

**Figure 1 healthcare-12-00293-f001:**
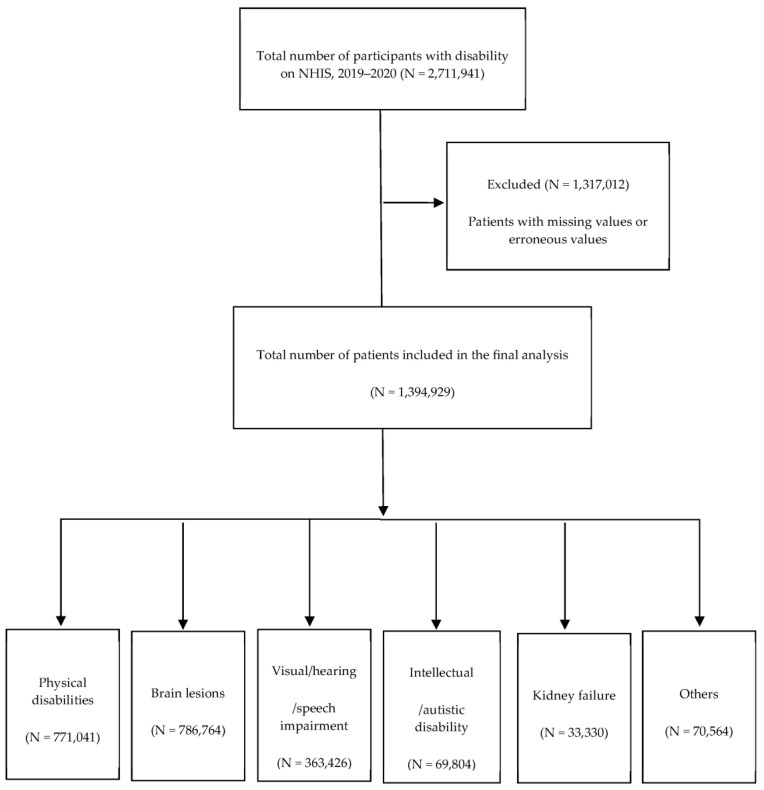
Flowchart of the patient selection process. NHIS, National Health Insurance Service. “Others” included speech-language disorder, psychiatric disability, facial disorder, epilepsy and internal organ impairments (heart, liver, and respiratory, intestinal and urinary organs).

**Table 1 healthcare-12-00293-t001:** General characteristics of the participants stratified by residential area and income levels.

Unit: Mortality								
		Seoul Metropolitan Area			Non-Metropolitan Area			Total
		High Income (%)	Median Income (%)	Low Income (%)	High Income (%)	Median Income (%)	Low Income (%)	
Disability type	Physical	1577 (39.3)	1293 (42.5)	1594 (39.1)	3550 (42.2)	2329 (43.5)	3196 (38.8)	13,539
	Brain lesions	598 (14.9)	387 (12.7)	538 (13.2)	1192 (14.2)	652 (12.2)	1088 (13.2)	4455
	Visual/hearing/speech	1279 (31.9)	885 (29.1)	1052 (25.8)	2877 (34.2)	1743 (32.5)	2262 (27.4)	10,098
	Developmental	28 (0.7)	30 (1.0)	148 (3.6)	36 (0.4)	46 (0.9)	443 (5.4)	731
	Kidney	308 (7.7)	270 (8.9)	298 (7.3)	417 (5.0)	310 (5.8)	477 (5.8)	2080
	Others	223 (5.6)	179 (5.9)	446 (10.9)	346 (4.1)	275 (5.1)	780 (9.5)	2249
Age, years	20–29	2 (0.0)	4 (0.1)	9 (0.2)	3 (0.0)	4 (0.1)	38 (0.5)	60
	30–39	8 (0.2)	22 (0.7)	29 (0.7)	8 (0.1)	26 (0.5)	64 (0.8)	157
	40–49	51 (1.3)	73 (2.4)	141 (3.5)	47 (0.6)	92 (1.7)	270 (3.3)	674
	50–59	163 (4.1)	235 (7.7)	477 (11.7)	209 (2.5)	336 (6.3)	937 (11.4)	2357
	60–69	495 (12.3)	660 (21.7)	1005 (24.7)	810 (9.6)	985 (18.4)	1700 (20.6)	5655
	70–79	1570 (39.1)	1014 (33.3)	1176 (28.9)	3031 (36.0)	1790 (33.4)	2180 (26.4)	10,761
	>80	1724 (43.0)	1036 (34.0)	1239 (30.4)	4310 (51.2)	2122 (39.6)	3057 (37.1)	13,488
Sex	Male	2712 (67.6)	2099 (69.0)	2659 (65.2)	5549 (65.9)	3707 (69.2)	5051 (61.3)	21,777
	Female	1301 (32.4)	945 (31.0)	1417 (34.8)	2869 (34.1)	1648 (30.8)	3195 (38.7)	11,375
BMI, kg/m^2^	Normal (18.5–25)	2408 (60.0)	1799 (59.1)	2344 (57.5)	5233 (62.2)	3337 (62.3)	4950 (60.0)	20,071
	Overweight (25–29)	1048 (26.1)	824 (27.1)	1048 (25.7)	1966 (23.4)	1276 (23.8)	1799 (21.8)	7961
	Obese (≥30)	174 (4.3)	123 (4.0)	199 (4.9)	286 (3.4)	207 (3.9)	381 (4.6)	1370
	Underweight (<18.5)	383 (9.5)	298 (9.8)	485 (11.9)	933 (11.1)	535 (10.0)	1116 (13.5)	3750
Smoking	Non-smoker	2456 (61.2)	1733 (56.9)	2371 (58.2)	5657 (67.2)	3316 (61.9)	5421 (65.7)	20,954
	Former smoker	1115 (27.8)	766 (25.2)	926 (22.7)	1885 (22.4)	1247 (23.3)	1311 (15.9)	7250
	Smoker	442 (11.0)	545 (17.9)	779 (19.1)	876 (10.4)	792 (14.8)	1514 (18.4)	4948
Drinking (number of days per week)	0	3271 (81.5)	2351 (77.2)	3249 (79.7)	7046 (83.7)	4297 (80.2)	6935 (84.1)	27,149
	1–2	439 (10.9)	394 (12.9)	461 (11.3)	658 (7.8)	528 (9.9)	692 (8.4)	3172
	3–4	143 (3.6)	153 (5.0)	187 (4.6)	291 (3.5)	232 (4.3)	285 (3.5)	1291
	≥5	160 (4.0)	146 (4.8)	179 (4.4)	423 (5.0)	298 (5.6)	334 (4.1)	1540
Total		4013 (100.0)	3044 (100.0)	4076 (100.0)	8418 (100.0)	5355 (100.0)	8246 (100.0)	33,152

BMI, body mass index.

**Table 2 healthcare-12-00293-t002:** Risk factors for mortality stratified by demographic characteristics.

		Crude HR	95% CI		*p*-Value
			Lower	Upper	
High income	Seoul metropolitan area	1	1	1	
	Non-metropolitan area	1.13	1.09	1.17	<0.0001
Median income	Seoul metropolitan area	1.21	1.15	1.27	<0.0001
	Non-metropolitan area	1.28	1.23	1.33	<0.0001
Low income	Seoul metropolitan area	1.22	1.16	1.27	<0.0001
	Non-metropolitan area	1.45	1.39	1.51	<0.0001
Age		1.07	1.07	1.07	<0.0001
Sex	Female	1	1	1	
	Male	1.68	1.64	1.73	<0.0001
BMI	Normal	1	1	1	
	Overweight	1.94	1.88	2.01	<0.0001
	Obese	1.01	0.99	1.04	0.3104
	Underweight	2.37	2.19	2.56	<0.0001
Smoking	Non-smoker	1.98	1.88	2.08	<0.0001
	Former smoker	2.14	2.05	2.24	<0.0001
	Smoker	1	1	1	
Drinking (number of days/week)	0	0.63	0.61	0.64	<0.0001
	1–2	0.66	0.63	0.70	<0.0001
	3–4	2.74	2.64	2.83	<0.0001
	5–7	1	1	1	
CCI		0.92	0.90	0.95	<0.0001

The multivariable Cox regression models were adjusted for age, sex, body mass index, smoking status, alcohol consumption, and Charlson comorbidity index. HR, hazard ratio; CI, confidence interval; BMI, body mass index; CCI, Charlson comorbidity index.

**Table 3 healthcare-12-00293-t003:** Risk of mortality by type of disability according to income and region.

			Crude HR (95% CI)	*p*-Value
Physical disability	High income	Seoul metropolitan area	1 (1–1)	
		Non-metropolitan area	1.16 (1.09–1.23)	<0.0001
	Median income	Seoul metropolitan area	1.26 (1.17–1.35)	<0.0001
		Non-metropolitan area	1.32 (1.24–1.41)	<0.0001
	Low income	Seoul metropolitan area	1.31 (1.22–1.41)	<0.0001
		Non-metropolitan area	1.53 (1.44–1.63)	<0.0001
Brain lesion	High income	Seoul metropolitan area	1 (1.00–1.00)	
		Non-metropolitan area	1.13 (1.02–1.25)	0.0154
	Median income	Seoul metropolitan area	1.10 (0.97–1.25)	0.1441
		Non-metropolitan area	1.14 (1.02–1.27)	0.0214
	Low income	Seoul metropolitan area	1.07 (0.95–1.20)	0.2852
		Non-metropolitan area	1.26 (1.14–1.39)	<0.0001
Visual/hearing/speech impairment	High income	Seoul metropolitan area	1 (1.00–1.00)	
		Non-metropolitan area	1.11 (1.04–1.18)	0.003
	Median income	Seoul metropolitan area	1.24 (1.13–1.35)	<0.0001
		Non-metropolitan area	1.33 (1.23–1.43)	<0.0001
	Low income	Seoul metropolitan area	1.11 (1.02–1.20)	0.015
		Non-metropolitan area	1.44 (1.34–1.54)	<0.0001
Autism	High income	Seoul metropolitan area	1 (1.00–1.00)	
		Non-metropolitan area	0.71 (0.44–1.17)	0.1798
	Median income	Seoul metropolitan area	0.96 (0.57–1.62)	0.873
		Non-metropolitan area	0.79 (0.50–1.27)	0.33
	Low income	Seoul metropolitan area	0.85 (0.57–1.28)	0.4476
		Non-metropolitan area	0.93 (0.63–1.37)	0.7221
Kidney failure	High income	Seoul metropolitan area	1 (1.00–1.00)	
		Non-metropolitan area	1.02 (0.88–1.18)	0.811
	Median income	Seoul metropolitan area	1.16 (0.98–1.36)	0.0819
		Non-metropolitan area	1.13 (0.97–1.33)	0.1261
	Low income	Seoul metropolitan area	1.083 (0.92–1.27)	0.3314
		Non-metropolitan area	1.393 (1.20–1.61)	<0.0001

The multivariable Cox regression models were adjusted for age, sex, body mass index, smoking status, alcohol consumption, and Charlson comorbidity index. HR, hazard ratio; CI, confidence interval.

## Data Availability

The raw data supporting this study’s results are available from the National Health Insurance System. However, limitations apply to data availability, as the data were used under license for this study and are not publicly available. Access to the original database was provided to the author upon reasonable request and with permission from the National Health Insurance System (https://nhiss.nhis.or.kr/bd/ab/bdaba000eng.do (accessed on 18 August 2021 to 8 August 2022).

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
