# Peer review of "Disability-Related Mortality Inequity in South Korea: Comparison in Terms of Seoul Metropolitan/Non-Metropolitan Areas and Income Levels"

_healthcare, 2024, doi:10.3390/healthcare12030293_

Round 1
Reviewer 1 Report
Comments and Suggestions for Authors
This study used an NHIS database to analyze demographic, diagnosis, and mortality information on persons with disability in South Korea. This study is important and reads well; however, I have some concerns/ recommendations for this paper that could improve it further:
1. Introduction – what are some examples of health disparities in people with disabilities?
2. Introduction – Lines 30 to 40 – I would use the definition of health inequity, which would be a better fit for things like race/SES than using ‘health inequality’. See https://dash.harvard.edu/bitstream/handle/1/41288111/51658%20v056p00647.pdf?sequence=1&isAllowed=y
3. The hypothesis seems weak – usually statistical hypothesis are like this: the null hypothesis was that there is no difference between mortality risk by disability status, residential area, or income level and the alternative hypothesis would be the converse.
4. Was a power analysis conducted?
5. The aim of the study is described but what was the original research question?
6. Line 90 has “771,04” this does not seem correct.
7. RE: Lines 104 to 105 “Seoul, Gyeonggi, and Incheon were defined as metropolitan areas, and the remaining cities were classified as non-metropolitan areas.” – why were some cities considered metropolitan and others not? I do not understand the logic. Were smaller cities considered less urbanized? If so, I think you need to state this clearly.
8. What were the theoretical frameworks that informed this study?
9. How was the analysis done, meaning which software was used?
10. Figure 1 has a lot of missing numbers and text is cut off – please consider revising and reducing the font size.
11. Line 117 says “The general characteristics of participants with disabilities who died” were presented in Table 1 – this seems problematic, what about those who lived?
12. Strengths and limitations – Good to see some of these were mentioned; what are other limitations such as lack of qualitative data that could have informed this type of quantitative study, for example?
13. What is novel here?
14. what are the implications of this research?
15. What future research is needed?
16. Re: Line 256-257 “Our results can help policymakers and researchers to determine ways to combat health inequalities.” But what about for people with disabilities? How does it help them since the study focuses on this group?
Thank you for the opportunity to review this manuscript.
Author Response
Thank you very much for taking the time to review this manuscript. Your comments were very insightful and have enabled us to improve the quality of our manuscript. We have incorporated changes that reflect the detailed suggestions you have graciously provided.
Please see the attachment.

Reviewer 2 Report
Comments and Suggestions for Authors
This is an interesting piece of work that brings into our attention the significant association of disability and inequalities in South Korea.
However there is a series of observations that I need to share, following the sequence of the text , in order to provide the authors with the opportunity to amend the manuscript before submitted again.
Introduction
"Economic growth has improved the overall level of health": this a big statement for which no consensus exists. I would avoid using it in the 1st sentence of my work.
"however, inequalities in socioeconomic aspects, such as individual income, residential area, type of disability, and lack of access to healthcare services, can lead to health inequality [1–7]. Health inequality" : replace the term inequalities in one of the three uses to avoid disturbing repetition.
"In South Korea, rapid economic growth and advancements in medical technology have led to a rapidly aging population and extended life expectancy." This needs to be underpinned by evidence.
"owing to the prolonged": please rephrase to "due to the prolonged"
"owing to high self-payments": the same as previously
"for people with disabilities is has also increased" this is not proper language use.
"Owing to many limitations": same as previously
"The income level of people with disabilities is relatively more marginalized from the benefits of medical welfare facilities" The use of English language is highly problematic impeding the meaning of the text. What does "the income level is more marginalized from benefits" might mean?
"The prevalence of physical disability is nearly five times greater among middle-aged adults with disabilities living in poverty than among those in high-income households". In the paragraph starting from line 5 you had started focusing on the financial aspect of disability and you suddenly transfer the argumentation to the epidemiological aspect, as the prevalence suggests. This is a problem of structure. In the following lines you move to co-morbidity referring to other health-related problems.
"Very few studies have investigated disability-related mortality risks [17]": I would argue that this is a highly disputable statement. The terms "excess mortality" and "disability" render thousands of results in academic search engines.
"However, the differences, based on socioeconomic and regional disparities, in the mortality rates among people with disabilities have not been studied yet." The same as above.
"can better function as": improper language use
A general comment: Your work is totally missing a theoretical perspective. What theoretical model do you use in respect to disability? There are three fundamental models that take an equal number of different and at some point opposing perspectives in analysing and interpreting disability. Which one do you adopt and why?
Material and methods
"The claims data of 2,711,941 patients" beyond the problematic use of grammar again, the number is not properly cited. It should be 2.711.941. This is the case for the subsequent numbers referring to individuals.
The Figure 1 is not cited properly, info is being missed from it! Look more cautiously in the boxes.
Discussion
"(high income/non-metropolitan areas: HR 166 = 1.13; median income/metropolitan areas: HR = 1.21; median income/non-metropolitan 167 areas: HR = 1.28; low income /metropolitan areas: HR = 1.22; low-income/non-metropolitan areas: HR = 1.45)": This piece of info does not suit to the Discussion. I suggest you erase. In this section you are expected to discuss and not to present your results, that should have already been presented previously.
"in many previous studies": language use
"they may be unable to participate in economic activities or may receive low salaries due to physical and mental problems": this is a disputable argument according to the social model of disability. To put it in explicit terms, it is not the problems of disabled persons that impedes their work and subsequently restricts their earnings, but the weakness of the social welfare to provide the conditions and facilities for them to participate in the productive process.
"We found that the distribution of people with disabilities between metropolitan and non-metropolitan areas was different": what does "different" mean? This is too vague for an academic text. You need to specify.
"their limited health conditions": this does not make any sense. Do you mean health resources? The reader should not be placed in a position to guess what you mean.
"people with disabilities were more likely to receive insufficient salaries and to have health disparities": they do not just "have health disparities", they experience health disparities
A general comment: This section lacks structure, you repeat your arguments in different points, making circles around the same idea without proceeding further your interpretations. I would suggest to rewrite the whole of the section following a clear and robust way to discuss your findings.
Several mistakes have been identified in the references lists.
e.g.:
Why did you abbreviate the title of the following reference?
1. Copeland, A.; Bambra, C.; Nylén, L.; et al. All in it together? The eff recession popul health Ineq Engl Sweden. Int J Health Serv. 1991– 292 2010, 45, 3-24:2015.
The following reference is outdated. Please find more up-to-date evidence to support your arguments.
4. Fukuda, Y.; Nakao, H.; Yahata, Y.; Imai, H. Are health inequalities increasing in Japan? The trends of 1955 to 2000. BioSci Trends. 298 2007, 1, 38–42.
The following is accessed on December 2021! We are two years later, it has to be updated!
7. Brennan Ramirez, L.K.; Baker, E.A.; Metzler, M. Promoting health equity; a resource to help communities address social determinants of health. Available online: https://www.cdc.gov/nccdphp/dch/programs/healthycommunitiesprogram/tools/pdf/SDOH-workbook.pdf (accessed on Dec 29 2021); Centers for Disease Control and Prevention United States Department of Health and Human Services, 2008.
Comments on the Quality of English Language
I have come across with several language errors, related to syntax, grammar but also the conventions of academic writing. The manuscript definitely needs extensive proofreading by a native English speaker.
Author Response

(The authors gave the same response as above.)

Reviewer 3 Report
Comments and Suggestions for Authors
1. The description of the NHIS database is rather general. It would be useful to provide more detailed information on the structure of the database, the specific variables used, and how data confidentiality is ensured.
2. Several variables are mentioned, such as income, area of residence, and type of disability, but the section lacks detail on exactly how these variables were measured and how they were classified.
3. The exclusion of 1,317,012 patients with missing or erroneous values is mentioned, but no detailed information is provided on the specific criteria used for exclusion.
4. The section mentions coding patients using a disability code, but no information is provided on how these codes were assigned and whether there were any specific criteria for classification.
5. Several control variables are mentioned, but no clear justification is provided as to why these specific variables were selected and how they contribute to the analysis.
6. Justify the reason for including "Kidney Failure" and also indicate which items are included in the Orhers category.
7. Include major limitations of the study and the implications of these for the study.
Comments on the Quality of English LanguageMinor editing of English language required.
Author Response

(The authors gave the same response as above.)

Round 2
Reviewer 1 Report
Comments and Suggestions for Authors
The authors have addressed all my concerns.
Thank you.
Reviewer 2 Report
Comments and Suggestions for Authors
The authors have revised the manuscript thoroughly considering carefully the feedback by the reviewers. The second version is substantially amended.
Reviewer 3 Report
Comments and Suggestions for Authors
None.
Comments on the Quality of English LanguageNone.